# Assessing changes in frailty status in an elderly population: Analysis of results from the SFGE tool

Edoardo Carnevale[ID]◉*, Michele Bisogno◉, Edoardo Trebbi, Clara Donnoli, Fausto Ciccacci, Fabio Riccardi, Paola Scarcella, Giuseppe Liotta[ID]

Department of Biomedicine and Prevention, University of Rome "Tor Vergata", Rome, Italy

◉ These authors contributed equally to this work
* edoardo.carnevale@students.uniroma2.eu

## Abstract

This pilot study explores the relationship between frailty and public health in older adults, evaluating the impact of an Individualized Care Plan (ICP) on frailty status over time. The study involved 125 individuals aged 65 and over recruited through the "Roma Tor Vergata" reference site for healthy and active aging. The Short Functional Geriatric Evaluation (SFGE) questionnaire was used to assess participants' frailty status at T0 (baseline) and T1 (six-month follow-up). The SFGE evaluates multidimensional frailty, covering psychophysical and socioeconomic domains. Older adults are classified according to four frailty levels: robust, pre-frail, frail, and very frail. Participants identified as pre-frail, frail, or very frail at T0 were also administered the "Sunfrail+" test, which further investigates the bio-psycho-social sphere. An ICP based on multidimensional assessments, was developed for each patient: it was shared with their general practitioner, and recommending specialist visits or rehabilitative interventions. 32% of participants experienced a change in frailty class during the six-month study period. 15.2% improved, 16.8% worsened, and 68% showed no change. Improvement is more frequent in pre-frail individuals, while worsening is predominantly found in robust individuals. Analysis reveals a significant correlation between social and psychological dimensions and the improvement/ worsening of frailty status. Participants who showed improvement reported a greater social activity and improved psychological well-being, while those who worsened reported a reduction in social activities and a decline in socioeconomic and psychological conditions. The study shows that the SFGE is a valuable tool for identifying changes in frailty status and that social networks and psychological well-being are significant factors for frailty status in older adults. Interventions aimed at strengthening social networks of older adults may be useful in preventing or delaying the onset of frailty.

**Data availability statement:** Data are available as supporting information.

**Funding:** The author(s) received no specific funding for this work.

**Competing interests:** The authors have declared that no competing interests exist.

## Introduction

The European Union's 2021 Aging Report indicates a significant shift in the European population's demographic profile over the coming decades [1]. The progressive aging will profoundly impact the planning and delivery of health and social services, leading to increased costs due to the rising demand for care services and the complexity of healthcare needs for the over-65 population. Population aging is a major challenge for global healthcare systems. The proportion of individuals experiencing frailty is growing because of the increase in life expectancy [2]. Frailty is characterized by multidimensional vulnerability and an elevated risk of adverse health events. To mitigate the strain on healthcare systems, new approaches are necessary, promoting healthy and active aging and preventing disability [3,4].

One of the most problematic aspects of population aging, potentially contributing to disability, is the condition of bio-psycho-social frailty [5,6]. This dynamic state of vulnerability is particularly prevalent in individuals over 65 years of age. It is marked by weakness and reduced physiological reserves, exposing individuals to an increased risk of diminished quality of life, falls, institutionalization, disability, and mortality. The literature emphasizes that frailty, especially in its early stages, is not irreversible. It is widely acknowledged that integrated interventions are crucial for slowing down or reversing the progression towards functional decline [7,8].

Factors contributing to increased vulnerability include genetic predisposition, family history, socio-environmental factors, early-life stress, and chronic diseases. Treatments for these conditions often contribute to vulnerability to psychiatric disorders, such as major depression or anxiety disorders. Conversely, factors enhancing resilience against stress include positive emotions, socio-environmental factors, cognitive flexibility, and physical exercise [9]. Frailty is a dynamic and multifactorial condition which necessitates an integrated and personalized approach from healthcare systems [10,11].

This study explores the intricate relationship between frailty and public health, analyzing its implications for the design and implementation of preventive and care interventions. This pilot study aims to assess bio-psycho-social frailty in individuals over 65, particularly identifying factors that may significantly influence frailty over approximately six months.

## Materials and methods

The "Rome Tor Vergata" Reference Site (RS) for Healthy and Active Aging is a consortium comprising a quadruple helix: the University of Rome Tor Vergata, the Tor Vergata Polyclinic, the Community of Sant'Egidio, and the IT services company We-COM. It joined the European Reference Site Collaborative Network (RSCN) due to the commitment and activities undertaken by its founding members in the field of prevention and promotion of health in the elderly population. Prevention days were organized, inviting individuals over 65 affiliated with the local public health unit (ASL RM1) to participate. Recruitment was on a voluntary basis after a publicity campaign. The patients arrived independently or accompanied to the S. Gallicano Hospital,

where assessments were conducted. Evaluations were also carried out at social centers in a different town area. Participants were selected through convenience sampling, including all elderly individuals who were present and willing to participate.

The study design was a quasi-experimental pre-post design to evaluate changes in frailty status after 6 months, using the SFGE questionnaire.

Recruitment of the patients took place from May 2023 to October 2024.

The SFGE questionnaire is a validated tool for the multidimensional assessment of frailty in older adults. This 16-item questionnaire assesses multidimensional frailty in community-dwelling older adults, encompassing the psychophysical and socioeconomic domains [12,13]. The SFGE questionnaire is used to assess the health status and well-being of older adults. The items are divided into several sections, investigating various aspects of an older person's life [14]. The SFGE questionnaire comprises sixteen items that evaluate different dimensions of frailty:

• **Item 1 – Age:** the questionnaire starts by asking the age, divided into three ranges: < 75, 75–85, > 85.

• **Item 2 – Education:** the education level is requested, distinguishing between primary/middle school and high school diploma/degree.

• **Item 3 – Cohabitants:** it investigates the subject's living situation, asking whether he/she lives alone, with a spouse, with a paid assistant, or with others. If with a spouse, the spouse's age is also requested.

• **Item 4 – Social Network:** it assesses the subject's social support, asking if there is someone who would help him/her in case of need and if he/she is involved in social activities or an association.

• **Item 5 and 6 – Assistance:** it investigates whether the subject receives assistance from municipal or local public health unit (ASL) services.

• **Item 7 and 8 – Financial Situation:** it asks whether the subject can make ends meet with his/her pension and, if not, which problems he/she had with in the last month.

• **Item 9 – Energy and Motivation:** it assesses the subject's level of energy and motivation.

• **Item 10–13 – Functional Abilities:** it assesses different functional abilities, asking whether the subject is able to bathe or shower on their own, to go out and if he/she is bedridden or confused.

• **Item 14 – Medication Intake:** it asks how many medicines the subject takes daily.

• **Item 15 – Falls:** it investigates whether the subject has fallen in the previous 3 months.

• **Item 16 – Hospital Admissions:** it asks whether the subject has been hospitalized or gone to the emergency room in the previous 3 months.

The questionnaire can be administered by healthcare professionals or non-healthcare professionals with a high school diploma. The average administration time is about 10 minutes. The total score is used to stratify older adults according to four levels: robust (score ≤ 0), pre-frail (1–2), frail (3–9), and very frail (≥10).

The inclusion criteria for the study were age over 65 and non-institutionalized status.

After obtaining informed consent, the SFGE questionnaire was administered at time 0 (T0) by specialist trainees and doctoral students from the School of Hygiene and Preventive Medicine of the University of Rome Tor Vergata. Subsequently, patients identified as pre-frail, frail, or very frail based on the "SFGE" questionnaire were administered the second-level "Sunfrail+" test. This is a 9-item questionnaire designed to investigate the individual's bio-psycho-social sphere. A positive response to one or more of the nine items triggers an alert that is further investigated with in-depth tests [15,16].

The Sunfrail+ questionnaire generates alerts related to several key factors known to significantly influence frailty:

- **Polypharmacy:** The use of multiple medications can lead to adverse drug reactions, cognitive impairment, and increased risk of falls, all of which contribute to frailty.

- **Nutrition:** Inadequate nutrition, including malnutrition and deficiencies in essential vitamins and minerals, weakens the body's resilience and accelerates the frailty process.

- **Fall Risk:** A history of falls or identified risk factors for falls (e.g., balance problems, muscle weakness) is a major indicator of frailty and can lead to injuries, disability, and reduced mobility.

- **Cognitive Decline:** Cognitive impairment, ranging from mild cognitive impairment to dementia, is closely linked to frailty, affecting an individual's ability to care for themselves and increasing their vulnerability.

- **Social Isolation:** Lack of social support and engagement is a potent predictor of frailty, contributing to psychological distress, reduced physical activity, and poorer overall health outcomes.

Individualized Care Plans (ICPs) were subsequently developed, addressing the specific needs of each participant based on the Sunfrail+ alerts and the overall multidimensional assessment from the SFGE. These ICPs focused on integrated interventions in the following key areas:

- **Nutritional support and counseling:** Participants with nutritional deficits or at risk of malnutrition received dietary advice, nutritional supplementation recommendations, and support to improve their eating habits.

- **Physical activity programs to prevent falls:** Interventions included tailored exercise programs designed to improve balance, strength, and mobility, thereby reducing the risk of falls and maintaining physical function.

- **Interventions aimed at enhancing social support:** To combat social isolation and promote social engagement, we collaborated with nonprofit entities like 'Long Live the Elderly!' [17] or social centres accessed by the clients enrolled in the study. This collaboration facilitated access to programs and services aimed at increasing social interaction, providing emotional support, and connecting older adults with community resources.

The Individual Care Plan (ICP) was developed for each patient, to be shared with the GP, recommending also specialist visits or rehabilitation interventions. After the administration of the questionnaires at T0, patients were called back for a follow-up at T1 after approximately 6 months. At T1, the T0 procedure was repeated, administering the same questionnaires, and any changes in frailty class were evaluated.

This multicenter study received ethical approval from the Ethics Committee of the Federico II University-A. Cardarelli National Hospital (protocol number N. 28422), which served as the central ethics committee. The study protocol was also submitted to the independent Ethics Committee of the Policlinico Tor Vergata on March 31, 2023, and was subsequently assigned protocol number 70.23. Written informed consent was obtained from every participant prior to their enrollment in the study, covering both their voluntary participation and the privacy of their data.

## Results

All 125 individuals completed the six-month follow-up (T1) out of the 125 over 65 subjects recruited and administered the SFGE questionnaire (S2 Dataset). The sample consisted of 40 men (32.26%) and 84 women (67.74%). The average age was 78 years, with 62 participants (49.5%) under 75 years old, 38 (30.3%) between 75 and 85 years old, and 24 (19.2%) over 85 years old.

Analysing the initial distribution of the sample (n = 125) based on frailty (robust, pre-frail, frail, very frail) (Table 1), it was observed that:

- 61.6% of the subjects were classified as Robust (Class 1).

- 24% were classified as Pre-frail (Class 2).

**Table 1. Sample distribution in frailty class.**

| CLASS T0 | N. patients | Percentage |
|---|---|---|
| 1 – Robust | 77 | 61,60% |
| 2 – Pre-frail | 30 | 24% |
| 3 – Frail | 16 | 12,80% |
| 4 – Very Frail | 2 | 1,60% |
| Total | 125 | 100% |

• 12.80% were classified as Frail (Class 3).

• 1.6% were classified as Very Frail (Class 4).

At T1, 40 participants (32%) demonstrated a change in class, indicating a shift in frailty status. 19 participants (15.2%) experienced improvement, 21 participants (16.8%) experienced worsening, while 85 participants (68%) showed no change in their frailty class. (Table 2)

Of the 125 subjects categorized based on their initial status (robust, pre-frail, frail, very frail) using the SFGE, it was observed that class improvement was more frequent in pre-frail individuals, 11 out of 19 improved (57.9%), whereas worsening was predominantly found in robust individuals, 15 out of 21 worsened (71,4%). (Table 3)

The responses with the highest frequency of variation between the two assessments were item 4 (formal or informal social network) (Table 4 and 5) and item 9 (psychological conditions) (Table 6 and 7). Notably, among those who improved, 42% (8 out of 19) showed improvement in response 4, and 52% (10 out of 19) showed improvement in response 9. Conversely, among those who worsened, 38% (8 out of 21) exhibited worsening in response 4, as well as 38% (8 out of 21) in response 9.

**Table 2. Variations at follow-up.**

| VARIATIONS AT T1 | | | | |
|---|---|---|---|---|
| CLASS T0 | Improved | Stable | Worsened | Total |
| 1 – Robust | 0 | 62 | 15 | 77 |
| 2 – Pre-frail | 11 | 14 | 5 | 30 |
| 3 – Frail | 6 | 9 | 1 | 16 |
| 4 – Very Frail | 2 | 0 | 0 | 2 |
| Total | 19 | 85 | 21 | 125 |

**Table 3. Contingency table between T0 and T1.**

| | CLASS T1 | | | | |
|---|---|---|---|---|---|
| CLASS T0 | 1 – Robust | 2 – Pre-frail | 3 – Frail | 4 – Very Frail | Total |
| 1 – Robust | 62 | 6 | 9 | 0 | 77 |
| 2 – Pre-frail | 11 | 14 | 5 | 0 | 30 |
| 3 – Frail | 5 | 1 | 9 | 1 | 16 |
| 4 – Very Frail | 1 | 0 | 1 | 0 | 2 |
| Total | 79 | 22 | 23 | 1 | 125 |

**Table 4. Variations Item 4 in Improved.**

**Item 4- Can you count on someone in case of need?**

| | | SFGE_4_t1 | | | Total |
|---|---|---|---|---|---|
| | | Yes, always | Sometimes | No | |
| SFGE_4_t0 | Yes, always | 6 | 1 | 1 | 8 |
| | Sometimes | 7 | 2 | 0 | 9 |
| | No | 1 | 0 | 1 | 2 |
| Total | | 14 | 3 | 2 | 19 |

**Table 5. Variations Item 4 in Worsened.**

**Item 4- Can you count on someone in case of need?**

| | | SFGE_4_t1 | | | Total |
|---|---|---|---|---|---|
| | | Yes, always | Sometimes | No | |
| SFGE_4_t0 | Yes, always | 8 | 4 | 3 | 15 |
| | Sometimes | 2 | 2 | 1 | 5 |
| | No | 0 | 0 | 1 | 1 |
| Total | | 10 | 6 | 5 | 21 |

**Table 6. Variations Item 9 in Improved.**

**Item 9 – Motivational condition**

| | | SFGE_9_t1 | | Total |
|---|---|---|---|---|
| | | Normal | Hypoactive/hyperactive | |
| SFGE_9_t0 | Normal | 6 | 1 | 7 |
| | Hypoactive/hyperactive | 10 | 2 | 12 |
| Total | | 16 | 3 | 19 |

**Table 7. Variations Item 9 in Worsened.**

**Item 9 – Motivational condition**

| | | SFGE_9_t1 | | Total |
|---|---|---|---|---|
| | | Normal | Hypoactive/hyperactive | |
| SFGE_9_t0 | Normal | 9 | 8 | 17 |
| | Hypoactive/hyperactive | 1 | 3 | 4 |
| Total | | 10 | 11 | 21 |

Using the Kruskal-Wallis test for the analysis of the distribution of responses 4 and 9 at T1 in the categories of improved and worsened, shows a statistically significant difference in scores ($p < 0.008$). This leads to a positive or negative change in frailty class. (Table 8)(Fig 1–2)

The other items analyzed showed no significant variations. (S1 Tables)

## Discussion

The results obtained from the application of the SFGE highlight a complex and nuanced picture of the bio-psycho-social frailty of the older adults. Assessment of Multidimensional frailty is a well-known approach to evaluate the risk of

**Table 8. Kruskal-Wallis Test on SPSS (Fig 1–2).**

| | Null hypothesis | Test | Sign. | Decision |
|---|---|---|---|---|
| 1 | The distribution of SFGE_4_t1 is the same across the categories of Variations. | Kruskal-Wallis test for independent samples | 0,003 | Reject the null hypothesis |
| 2 | The distribution of SFGE_9_t1 is the same across the categories of Variations. | Kruskal-Wallis test for independent samples | 0,007 | Reject the null hypothesis |

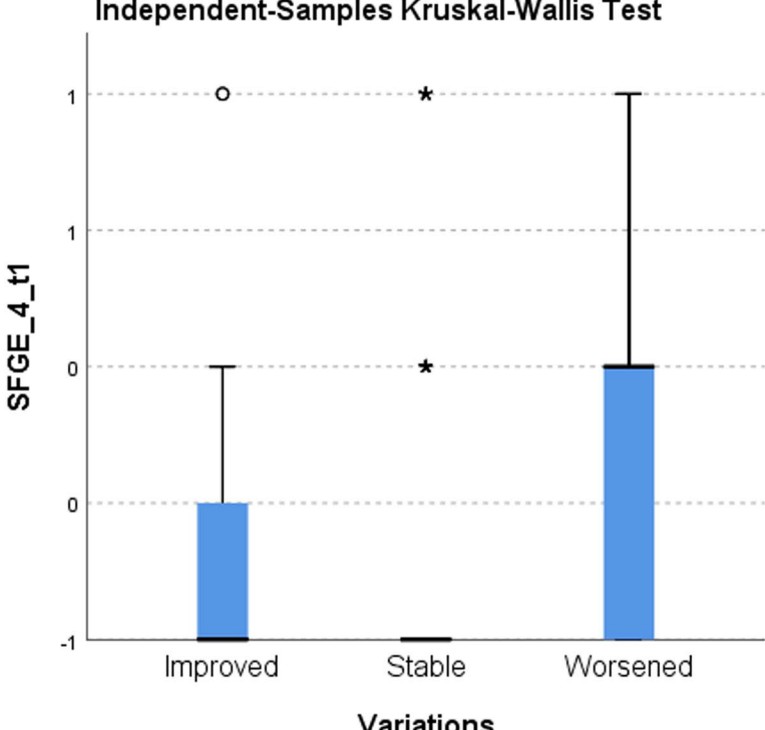

**Fig 1. Kruskal-Wallis test for independent samples on SFGE_4_t1.**

negative events connected to a specific condition [18]. However, a general approach to Bio-psycho-social, not linked to a specific diseases could also provide valuable information at population level in order to provide the scientific background to intervention addressing transversal risk factors as social isolation, polypharmacy, malnutrition, risk of falls or cognitive decline.

The social dimension reveals a slight predominance of worsening, implying that some aspects of the older adults' social relationships were experiencing decline. It is essential to investigate which specific aspects of social relationships were affected and whether this is related to individual characteristics. The economic dimension is the least changed, with a majority of participants experiencing no changes. This could indicate that the participants' economic conditions were already stable. Results concerning the psychological dimension show slightly more improvements than deterioration. This suggests a potential for positive change in the older adults' psychological well-being. The physical dimension demonstrates stability across the functional ability items of the SFGE questionnaire (items 10–13). This is likely attributed to the short interval between the two evaluations.

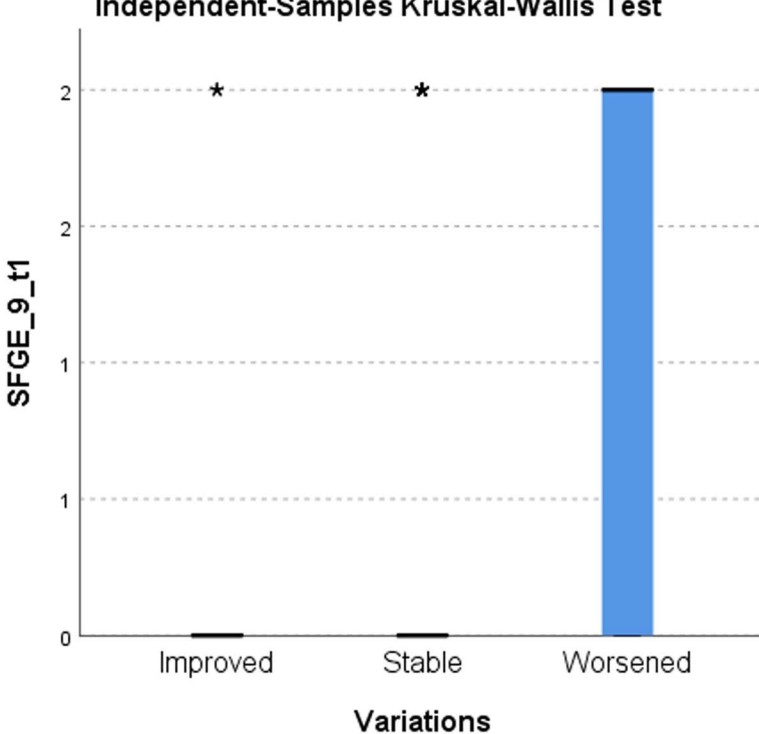

**Fig 2. Kruskal-Wallis test for independent samples on SFGE_9_t1.**

An aspect to be considered when evaluating changes in older adults frailty status is the follow-up duration. This study considered a period of approximately six months. While this timeframe allowed for detecting significant variations in some dimensions, it is possible that a longer follow-up would have revealed even more pronounced changes, particularly regarding the physical dimension and its implications for daily functionality. Furthermore, a longer follow-up could enable the observation of the long-term evolution of the intervention's effects and the identification of any delayed effects. This study's findings are also limited by its small sample size and the reliance on a non-random sampling technique. This sampling, while practical for this pilot study, introduces the possibility of a selection bias. Participants who volunteer to participate in such studies may differ systematically from those who do not, potentially limiting the representativeness of our sample. For instance, they might be more health-conscious or have better access to healthcare services. The assessment of the psycho-physical status is necessarily coarse due to the characteristics of the tool, which is a first level assessment that should be carried out by non-health personnel. However, this kind of questionnaire is strongly indicated for population studies [19].

The data demonstrate that improvements are more frequent in the pre-frail group, and the primary factors associated with improvements are attributed to increased social activities, improved psychological well-being, and the presence of a reliable reference figure. Conversely, deterioration is more prevalent in the robust group, with the main associated factors being a reduction in social activities, consequently increasing the risk of social isolation, and a decline in socioeconomic and psychological conditions.

The results indicate that social networks and psychological well-being are significant factors influencing the frailty status of older adults. Moreover, the SFGE has proven to be a valuable tool for identifying changes in frailty status, particularly within the pre-frail class, where they occur more frequently.

## Conclusions

The SFGE underscores the crucial role of the social domain in determining the risk of negative health outcomes for older adults. Early identification of frailty is essential to implement personalized interventions and prevent functional decline and complications. Repeated administration of the SFGE (e.g., at T0 and T1) allows to assess the effectiveness of interventions, and to adapt the ICP, and identifying older adults requiring more intensive support. The SFGE can help guide frail older adults toward specialized pathways tailored to their individual needs. Interventions aimed at strengthening older adults' social networks can be beneficial in preventing or delaying the onset of frailty. A robust social network, as indicated by a positive response to question 4, positively impacts frailty status by providing a sense of belonging and connection, thereby reducing the risk of loneliness and social isolation. The social network contributes to improving the living conditions of older adults by potentially offering practical help with daily activities, such as shopping, meal preparation, housework, or transportation. This concrete support can enhance the autonomy and quality of life of older adults and empower them to make informed decisions about their health and well-being.

## Supporting information

**S1 Tables.** XXX.
(XLSX)

**S2 Dataset.** XXX.
(XLSX)

## Author contributions

**Conceptualization:** Clara Donnoli, Fausto Ciccacci.

**Data curation:** EDOARDO CARNEVALE, Michele Bisogno.

**Investigation:** EDOARDO CARNEVALE.

**Project administration:** Giuseppe Liotta.

**Supervision:** Fausto Ciccacci, Paola Scarcella.

**Writing – original draft:** EDOARDO CARNEVALE, Michele Bisogno, Edoardo Trebbi, Clara Donnoli.

**Writing – review & editing:** Fabio Riccardi, Giuseppe Liotta.

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
