## [Decision Letter · Decision Letter 0]

Dear Dr. CARNEVALE,

Thank you for submitting your manuscript to PLOS ONE. After careful consideration, we feel that it has merit but does not fully meet PLOS ONE’s publication criteria as it currently stands. Therefore, we invite you to submit a revised version of the manuscript that addresses the points raised during the review process.

We look forward to receiving your revised manuscript.

Kind regards,

Pasquale Abete

Academic Editor

PLOS ONE

Journal Requirements:

4. In the online submission form, you indicated that data cannot be shared publicly. Data are available on request from the corresponding author.. 

Additional Editor Comments:

According to Reviewers' decision, the manuscript needs a major revision.

Reviewers' comments:

Reviewer's Responses to Questions

**Comments to the Author**

1. Is the manuscript technically sound, and do the data support the conclusions?

Reviewer #1: Yes

Reviewer #2: Partly

2. Has the statistical analysis been performed appropriately and rigorously?

Reviewer #1: Yes

Reviewer #2: Yes

3. Have the authors made all data underlying the findings in their manuscript fully available?

Reviewer #1: Yes

Reviewer #2: Yes

4. Is the manuscript presented in an intelligible fashion and written in standard English?

Reviewer #1: Yes

Reviewer #2: Yes

Reviewer #1: The study evaluate the relationship between frailty and public health in older

adults, evaluating the impact of an Individualized Care Plan (ICP) on frailty status over time. The study involved 125 individuals aged 65 and over recruited through the "Roma Tor Vergata" reference site for healthy and active aging in Italy. The Short Functional Geriatric Evaluation (SFGE) questionnaire was used to assess participants' frailty status at baseline and after six-month of follow-up. The SFGE evaluates multidimensional frailty, covering psychophysical and socioeconomic domains. Older adults are classified according to four frailty levels: robust, pre-frail, frail, and very frail. Participants identified as pre-frail, frail, or very frail at baseline were also administered the "Sunfrail+" test, which further investigates the bio-psycho-social sphere. An ICP based on multidimensional assessments, was developed for each patient: it was shared with their general practitioner, and recommending specialist visits or rehabilitative interventions. 32% of participants experienced a change in frailty class during the six month study period. 15.2% improved, 16.8% worsened, and 68% showed no change. Improvement is more frequent in pre-frail individuals, while worsening is predominantly found in robust individuals. Analysis reveals a significant correlation between social and psychological dimensions and the improvement/ worsening of frailty status. Participants who showed improvement reported a greater social activity and improved psychological well-being, while those who worsened reported a reduction in social activities and a decline in socioeconomic and psychological conditions.

I found the study of interest, the methodology is correct and data support conclusions. The main problem of this study is the small sample of subjects enrolled. As Pilot study I hope that these results will be confirmed with a larger study conducted not only in one restricted area but all over Italy and Europe. Social networks and psychological well-being are really different among regions and nations, and therefore interventions aimed at strengthening social networks of older adults maycould be different in preventing or delaying the onset of frailty in different nations.

Reviewer #2: This manuscript titled "Assessing Changes in Frailty Status in an Elderly Population: Analysis of Results from the SFGE Tool" explores the impact of an Individualized Care Plan on frailty status over time in older adults. Using the SFGE questionnaire, the study assesses participants' frailty status at the beginning of the study and six months later. The results indicate that 32% of the participants experienced a change in their frailty class, with some improving and others worsening, primarily influenced by their social activities and psychological well-being.

Overall, the manuscript offers valuable insights into the dynamics of frailty among the elderly, emphasizing the crucial role of social networks in influencing frailty outcomes. It suggests that targeted interventions designed to enhance social connections could be instrumental in preventing or delaying the onset of frailty. However, the manuscript would benefit significantly from revisions addressing various issues that currently detract from its overall impact and readability.

- One major concern is the ambiguous definition of the "Individualized Care Plan." The inherent uniqueness of each plan, tailored to patient-specific needs, complicates direct comparisons and limits the interpretability of the results. The manuscript lacks a description of these care plans, especially regarding whether they address the key determinants of social frailty. The issue is that the study aims to analyze changes in frailty following interventions that are not adequately described in the manuscript, which limits the interpretability of the results. If social determinants are the primary drivers of worsening frailty, it is essential to know whether the Individualized Care Plan included measures to counteract social frailty. Further discussion on the adherence and execution of these plans is necessary to adequately assess their effectiveness.

- Additionally, while the clarity of the statistical analysis is commendable, a complementary analysis exploring the specific items driving improvements in frailty would enhance understanding.

- It remains unclear how the "Sunfrail+" tool is incorporated into the study.

- The use of convenience sampling raises questions about the generalizability of the findings. A discussion on potential biases introduced by this method would strengthen the study's validity. Moreover, the SFGE, although a multidimensional tool, does not offer an objective evaluation of physical capacity. Addressing this limitation in the discussion could clarify how it might affect follow-up results.

- Expanding the discussion to compare the utility of multidimensional versus physical frailty models would be beneficial: please see and discuss “Testa G at al. Physical vs. multidimensional frailty in older adults with and without heart failure. ESC Heart Fail. 2020 Jun;7(3):1371-1380. doi: 10.1002/ehf2.12688”.

- Finally, the manuscript would benefit from a thorough language revision to correct occasional grammatical errors and improve the quality of English (for instance: "Recruitment of the patients take place from May 2023 to October 2024." should be "Recruitment of the patients takes place..."). Enhancing the graphical presentation of results could also help in effectively conveying the study's findings.

**Do you want your identity to be public for this peer review?** For information about this choice, including consent withdrawal, please see our Privacy Policy

Reviewer #1: No

Reviewer #2: No

---

## [Author Response · Author response to Decision Letter 1]

19 May 2025

In response to Reviewer #1's concerns about sample size, we have acknowledged this limitation and discussed plans for a larger multi-center study across different regions to better account for regional variations in social networks and psychological well-being.

For Reviewer #2, we have:

Provided a detailed description of the Individualized Care Plans, particularly clarifying how they address social determinants of frailty

Added complementary analysis identifying specific items driving frailty improvements

Clarified the integration of the Sunfrail+ tool in our methodology

Discussed the limitations of convenience sampling and potential biases

Acknowledged the SFGE's constraints in objective physical capacity evaluation

Added a comparative discussion of multidimensional versus physical frailty models, including reference to Testa et al.

Thoroughly revised the manuscript for grammatical errors and improved English expression.

We believe these revisions have substantially strengthened our manuscript and hope it now meets PLOS ONE's publication criteria.

Sincerely,

Dott. Edoardo Carnevale.

---

## [Editor Report · Decision Letter 1]

Assessing Changes in Frailty Status in an Elderly Population: Analysis of Results from the SFGE Tool

PONE-D-25-01358R1

Dear Dr. CARNEVALE,

We’re pleased to inform you that your manuscript has been judged scientifically suitable for publication and will be formally accepted for publication once it meets all outstanding technical requirements.

Kind regards,

Pasquale Abete

Academic Editor

PLOS ONE

Additional Editor Comments (optional):

No further comments
---

## [Editor Report · Acceptance letter]

PONE-D-25-01358R1

PLOS ONE

Dear Dr. CARNEVALE,

I'm pleased to inform you that your manuscript has been deemed suitable for publication in PLOS ONE. Congratulations! Your manuscript is now being handed over to our production team.

Kind regards,

on behalf of

Prof. Pasquale Abete

Academic Editor

PLOS ONE